# Perceptions of barriers and facilitators for cervical cancer screening from women and healthcare workers in Ghana: Applying the Dynamic Sustainability Framework

**Adwoa Bemah Boamah Mensah**[1]*, **Thomas Okpoti Konney**[2], **Ernest Adankwah**[3,4], **John Amuasi**[4,5], **Madalyn Nones**[6], **Joshua Okyere**[1,7], **Kwame Ofori Boadu**[8], **Felicia Maame Efua Eduah**[9], **Serena Xiong**[10], **Jeong Robin Moon**[11], **Beth Virnig**[12], **Shalini Kulasingam**[6]

**1** Department of Nursing, School of Nursing and Midwifery, Kwame Nkrumah University of Science and Technology, Kumasi, Ghana, **2** Department of Obstetrics and Gynecology, School of Medical Sciences, Kwame Nkrumah University of Science and Technology, Kumasi, Ghana, **3** Department of Medical Diagnostics, Faculty of Allied Health Sciences, Kwame Nkrumah University of Science and Technology, Kumasi, Ghana, **4** Global Health Department, School of Public Health, Kwame Nkrumah University of Science and Technology, Kumasi, Ghana, **5** Kumasi Center for Collaborative Research in Tropical Medicine, UPO PMB, KNUST, Kumasi, Ghana, **6** Division of Epidemiology and Community Health, School of Public Health, University of Minnesota, Minneapolis, Minnesota, United States of America, **7** Department of Population and Health, University of Cape Coast, University Post Office, Cape Coast, Ghana, **8** Ghana Health Service, Kumasi South Regional Hospital, Kumasi, Ashanti Region, Ghana, **9** Ghana Health Service, Suntreso Government Hospital, Kumasi, Ashanti Region, Ghana, **10** Division of Public Health Sciences, University of Washington, St. Louis, Missouri, United States of America, **11** Department of Health Policy and Management, Graduate School of Public Health and Health Policy, City University of New York, New York City, New York, United States of America, **12** College of Public Health and Health Professions, University of Florida, Gainesville, Florida, United States of America

* bbemahc2000@gmail.com

## Abstract

Cervical cancer screening has reduced cervical cancer-related mortality by over 70% in countries that have achieved high coverage. However, there are significant geographic disparities in access to screening. In Ghana, although cervical cancer is the second most common cancer in women, there is no national-level cervical cancer screening program, and only 2–4% of eligible Ghanaian women have ever been screened for cervical cancer. This study used an exploratory, sequential mixed-methods approach to examine barriers and facilitators to cervical cancer screening from women's and healthcare workers' perspectives. These were further informed by the Dynamic Sustainability Framework (DSF), in particular, two domains, namely the practice setting and ecological system. Two convenience samples of 215 women and 17 healthcare personnel were recruited for this study. All participants were from one of three selected clinics (Ejisu Government Hospital, Kumasi South Hospital, and the Suntreso Government Hospital) in the Ashanti region of Ghana. Descriptive analyses were used to group the data by practice setting. Statistical differences in means and proportions were used to evaluate women's barriers to cervical cancer screening.

**Data availability statement:** The datasets generated and/or analyzed during the current study is publicly available through the link https://figshare.com/articles/dataset/ Data_Set_for_Barriers_in_Cervical_cancer_Screening/27119364. DOI (10.6084/ m9.figshare.27119364)

**Funding:** The study was funded by the Institute of Global Cancer Prevention Research (IGCPR). Masonic Cancer Center, University of Minnesota [Grant # CPRC 2022LS031], (SK, ABBM, TOK, and BV). The funders had no role in study design, data collection and analysis, decision to publish, or preparation of the manuscript.

**Competing interests:** The authors have declared that no competing interests exist.

Quantitative findings from the women's survey informed qualitative, in-depth interviews with the healthcare workers and were analyzed using an inductive thematic analysis. The median age of women and healthcare workers was 37.0 years and 38.0 years, respectively. Most women (n = 194, 90.2%) reported never having been screened. Women who had not been screened were more likely to have no college or university education. Practice setting factors included long clinic wait times and distance to the clinic. Ecological system factors identified were population characteristics such as lack of knowledge about available services, shyness when undergoing a clinician-performed pelvic exam, and requiring a spouse's permission before scheduling. These findings highlight the need for non-clinician-based, culturally sensitive cervical cancer screening options such as self-collected HPV tests to increase screening participation in Ghana.

## Introduction

Cervical cancer is a leading cause of cancer deaths among women in low- and middle-income countries (LMIC), primarily due to limited access to screening [1]. Ghana has a population of 10.6 million women (approximately 63.2% of the female population) aged 15 years and older who are at risk of developing cervical cancer [2]. Additionally, cervical cancer is the second most frequently diagnosed cancer in Ghanaian women, with an age-standardized incidence rate of 27.4 per 100,000 women and an age-standardized mortality rate of 17.8 per 100,000 women. In contrast, despite a similar population size, Spain has age-standardized cervical cancer incidence and mortality rates of 8.2 and 1.7 per 100,000 women, respectively, due to the widespread availability of screening and treatment [3].

Ghana faces many barriers in its effort to address the high incidence of cervical cancer. Currently, there is no national human papillomavirus (HPV) vaccination campaign to protect women from contracting HPV, the leading cause of cervical cancer [4]. As a result, both vaccination and screening rates remain low [2]. The majority of women present to clinics with advanced-stage cervical cancer [5]. To address this, Ghana's National Reproductive Health Policy was revised in 2014 to integrate cervical cancer screening into existing reproductive health programs such as family planning and sexually transmitted infections management services [6,7]. The policy includes recommendations for screening using Visual Inspection with Acetic Acid (VIA) and Papanicolaou tests (pap smear) for women ages 25–45 years and cryotherapy for treating precancerous cervical lesions. Despite this policy, which has been in place for over nine years, only 2.4 to 4% of eligible Ghanaian women are screened annually for cervical cancer [8–10]. Interviews with women in Ghana who have been diagnosed with late-stage cervical cancer suggest high costs, a lack of knowledge, and a lack of access to screening facilities are important factors [11,12].

Until recently, the World Health Organization (WHO) has recommended cervical cytology and VIA for cervical cancer screening in LMICs. Of the two, VIA has been more readily adopted across sub-Saharan Africa (SSA) due to the availability

and low cost of acetic acid for visualizing cervical abnormalities, the low cost of training personnel to conduct VIA-based screening, and the ability to carry out immediate treatment if abnormal regions of the cervix are identified (referred to as 'see and treat'). However, limitations of this approach include low sensitivity, low reproducibility, and the need for trained personnel to conduct pelvic exams [13–16]. Although cytology has an improved sensitivity compared to VIA, it also involves a pelvic exam by trained clinic staff for specimen collection, is more expensive than VIA, and requires training to interpret laboratory results [16]. As a result, the WHO updated their cervical cancer screening guidelines in 2022 to primarily recommend HPV testing in conjunction with self-collected samples for women ages 30–50 years [17].

HPV-based screening has a significantly higher sensitivity compared to VIA and cytology for the detection of high-grade cervical lesions [18,19]. HPV testing can be performed with self-collected samples, which allows patients' privacy and, importantly, reduces reliance on clinics with trained personnel. While this approach to cervical cancer screening is appealing, its implementation in countries in SSA is not straightforward. Hurdles include determining when and where to offer screening, how to best instruct women on self-collection methods, which collection device and HPV test to use, and how to triage HPV-positive women. Deciding how to address these hurdles requires an understanding of the unique barriers and facilitators faced by a population operating within a specific healthcare setting in a given country.

### Conceptual framework

This study was guided by the Dynamic Sustainability Framework (DSF) developed in 2013 by Chambers and colleagues. According to Chambers et al. [20], the DSF is a framework designed to address the sustainability of health interventions over time. This framework comprises three components/domains: the intervention, practice setting, and broader ecological system. The intervention is the health programs that have been implemented to yield a desired health outcome. The practice setting reflects the local clinical and community context in which the intervention takes place; this includes but is not limited to "human and capital resources, information systems, organizational culture, climate and structure, and processes for training and supervision of staff" [20]. The ecological system defines the broader context within which care is delivered. It includes other practice settings, policies, and regulations, as well as population characteristics that influence the alignment between the intervention and practice settings. Chambers et al. [20] argue that achieving a desired health outcome is dependent on how well the intervention fits into the practice setting and ecological system and that the changing context of healthcare provision necessitates revisiting each of these elements to ensure the success of an intervention over time.

In the context of this study, the "intervention" component of the DSF refers to the existing cervical cancer screening program that has been implemented in three major health facilities in Kumasi, Ashanti region. The program, herein referred to as CERVICARE was established by the Ghana Health Service (GHS) in three hospitals to provide cervical cancer screening services for women in the region to reduce cervical cancer incidence and mortality. Services currently offered as part of the CERVICARE program include pap smear with cytology, VIA, and cryotherapy for treating precancerous cervical lesions. Study sites were established in one peri-urban clinic (Ejisu Government Hospital) as well as two of the three CERVICARE clinics: Kumasi South Regional Hospital (KSRH) and South Suntreso Government Hospital (SSGH), which established screening in 2009 and 2010, respectively. While programs at both hospitals have seen recent increases in the number of women attending the clinic (KSRH: 363 in 2019–653 in 2023; SSGH: 120 in 2020–1,159 in 2023), the concern is that most eligible women are not being screened. Practice-setting factors are those related to the immediate clinical or community setup, such as infrastructure, staff availability, and logistical constraints. Ecological system factors include broader contextual factors such as characteristics of the population, including knowledge level, socio-cultural norms, financial constraints, access to transportation or information, and local or national policies that may help or hinder care. We developed our study tools to examine the barriers and facilitators to cervical cancer screening from the perspectives of women (Study I) and, separately, healthcare workers (Study II) which reflect an adaptation of the practice setting domain of the DSF (Fig 1). Data analysis and interpretations were guided by the adapted DSF.

PLOS Global Public Health

**Fig 1. The adapted Dynamic Sustainability Framework as it was used within the content of this study.**

The conceptual framework begins with the intervention box at the top, represented by the current cervical cancer screening programs at KSRH and South Suntreso Government Hospital. An arrow connects the intervention and practice setting domains to illustrate how screening programs are implemented within clinical and community setups, as well as how programs are impacted by the availability of resources, logistics, and overall service delivery. Simultaneously, an arrow connects the ecological system and intervention domains to highlight how broader contexts, such as socioeconomic factors and cultural beliefs, influence the screening programs. The practice setting and ecological system domains are positioned side-by-side beneath the intervention domain, depicting the interaction between the immediate clinical/community environment and the broader context in which the intervention is delivered.

## Study I: Quantitative survey

### Materials and methods

**Ethics statement.** Approval for this study was provided by the Committee on Human Research, Publication and Ethics, Kwame Nkrumah University of Science and Technology, Ghana (CHRPE/AP/043/22) and the Cancer Center Protocol Review Committee, University of Minnesota, USA (CPRC# 2022LS045). The participants received written information about the study, their informed consent was sought, and participation was voluntary and anonymous.

**Study design and setting.** We adopted a cross-sectional design to investigate the barriers and facilitators of cervical cancer screening in women aged 30 years and over in Ghana. The study consisted of women presenting to two (2) urban facilities of the Ghana Health Services (GHS) CERVICARE program (Pap smear VIA, and cryotherapy for treating precancerous cervical lesions) and one peri-urban facility in the Ashanti Region of Ghana. These hospitals are Kumasi South Regional Hospital (KSRH), Suntreso Government Hospital (SGH), and Ejisu Government Hospital (EGH). The Ashanti region is the second most heavily populated region in Ghana, with a female population of 2,760,549 in 2021 [21]. SRH and SGH are urban facilities and together serve 65.8% of the total Kumasi population and 70.5% of the total population of women of fertility age (WIFA) in Kumasi. EGH is a peri-urban facility located in the municipality of Ejisu, which has a smaller population of approximately 181,000 individuals. We included the EGH because it has no cervical cancer screening facilities, and women served by this facility must either wait for outreach services or travel to KSRH

for cervical cancer screening. Also, women accessing care at the EGH are commonly referred to the KSRH for cervical cancer screening. Together, the selected facilities serve individuals from three metropolitan assemblies in the Ashanti region: Ejisu, Asokwa, and Kumasi. These areas were chosen because participation in screening is low [22].

**Sample and sampling procedure.** We estimated the sample size using the formula: $n = \frac{N \cdot Z2 \cdot p \cdot (1-p)}{(N-1) \cdot e2 + Z2 \cdot p \cdot (1-p)}$ where population size (N) is 321 (based on 2021 attendee number from facilities), the Z-value (Z) for a 95% confidence level is 1.96, the population proportion (p) is conservatively estimated at 0.5, and the margin of error (e) is set at 0.05. We then calculated an attrition rate of 20%, bringing the estimated sample size to 210. We identified women through a convenience sampling approach. This approach was chosen to ensure that participants were readily available to take part in the study. Women who visited the health facility for care during the survey were considered eligible for participation. Healthcare providers assisted in identifying eligible women based on the study's inclusion criteria. Women were then approached when they visited the facility. With the support of healthcare providers, the research team intermittently approached women throughout the data collection period rather than selecting them consecutively on a limited number of days to capture a diverse range of participants who reflected the broader patient population attending the facility during that time.

**Recruitment and consent.** The study team introduced the study with an introductory letter sent to each administrator. Following agreement to participate and an ethical approval letter, a pre-data collection overview of the study was provided to the clinic management and at each hospital. Ethical approval for this study was provided by the Committee on Human Research, Publication, and Ethics, Kwame Nkrumah University of Science and Technology, Ghana (CHRPE/AP/043/22) and the Cancer Center Protocol Review Committee, University of Minnesota, USA (CPRC# 2022LS045). Eligible individuals for the study were women aged 30 years and older who were able to undergo cervical cancer screening, lived within the catchment area of one of the three hospitals, could provide informed consent, and could comprehend the survey questions, either verbally or in written form. Women initially found by research assistants (RAs) who met the eligibility criteria were informed about the study and invited to take part. If a woman expressed interest, she was escorted to a private room where the research assistant provided detailed information about the study and conducted a thorough eligibility assessment. Once eligibility was confirmed, and if the woman was willing to participate after receiving all necessary study details, written informed consent was obtained, and the questionnaire was administered. Recruitment of participants took place from August 1, 2022, to October 31, 2022.

**Data collection tool and procedure.** A survey was adapted from a pre-tested survey used in a study that evaluated barriers and facilitators to cervical cancer screening in a population of 900 women living in Uganda [23]. Similar surveys have assessed barriers and facilitators to screening uptake in other countries in Sub-Saharan Africa [24]. The questionnaire was developed for a larger study seeking to optimize HPV-based cervical cancer screening in Ghana and was drafted in English translated into Twi and back-translated into English to check for potential errors. Research assistants reviewed all the questions to ensure clarity. The data collection tool used for Study I included two sections: the first section collected demographic information, including the participant's age, education level, marital status, occupation, average income, number of dependents, and length of living in the current district. The second section focused on the ecological system and practice-setting domains of the DSF. Questions included population characteristics (ecological system) that influence willingness to be screened as well as aspects of the clinic (practice setting) that affect participation in cervical cancer screening were asked from the perspective of women. The questionnaire was programmed in REDCap (Research Electronic Data Capture) to allow real-time data collection and uploading and to also help with quality control as well as allow for efficient data processing [25,26].

**Statistical analysis.** All analyses were performed using R Statistical Software (v4.1.3; R Core Team 2021) [27]. Data cleaning included evaluating missing and non-response rates for various questions. All variables used for analysis had <1% of missing responses. Descriptive statistics were used to summarize population characteristics and evaluate other practice-setting constraints that affect participation in cervical cancer screening. Continuous variables

were quantified using the median and IQR while frequencies were used to quantify categorical variables. Additionally, other practice setting constraints were stratified by women's education level (some college or university versus no college or university), marital status (married or living together versus single, divorced or widowed) as well as rural versus urban.

Categorical variables were created based on prior studies examining socio-demographic variables that predict Pap-smear uptake in Ghanaian women. For marital status, women who were married or living with a partner were compared to women who were single, divorced, widowed, or separated. For educational status, women who had some college or university education were compared to women with secondary school education or less. Women living in rural areas were also compared to those living in urban areas, which has previously been shown to influence the uptake of cervical cancer screening [28]. After stratifying by demographic variables, prevalence ratios were used to assess differences in response frequency by demographic category, and two-sided t-tests were used to analyze the difference in means and differences in proportions for continuous variables. All statistical tests were deemed significant at the α=0.05 level.

**Results.** A total of 215 women were surveyed. Characteristics of this population are presented in Table 1. The median age of women was 37.0 years (IQR: 32.0 - 45.5). Women had varying levels of educational attainment. Most women were married (69.3%, n = 149) and self-employed (62.8%, n = 135). The median number of dependents was four.

**Table 1. Demographics for women participants.**

| Participant Characteristic (n = 215) | |
|---|---|
| Age in years* | 37.0 (32.0–45.5) |
| Education, n (%) | |
| No primary school | 12 (5.6) |
| Primary school | 22 (10.2) |
| Junior high school | 78 (36.3) |
| Senior high school | 53 (24.7) |
| Some college or university | 50 (23.2) |
| Marital Status, n (%) | |
| Single | 23 (10.7) |
| Married | 149 (69.3) |
| Widowed | 18 (8.4) |
| Divorced/Separated | 12 (5.6) |
| Living with a partner | 13 (6.0) |
| Occupation, n (%) | |
| Agriculture/Farming | 12 (5.6) |
| Self-employed | 135 (62.8) |
| Housewife | 12 (5.6) |
| Civil/Government/Private Sector Employee | 40 (18.6) |
| Other | 16 (7.4) |
| Dependent | |
| Number of Dependents* | 4.0 (3.0–5.5) |
| Length Living in District, n (%) | |
| All of my life | 19 (8.8) |
| Less than 1 year | 22 (10.2) |
| Between 1 and 5 years | 74 (34.4) |
| More than 5 years | 100 (46.5) |

*Age and number of dependents are reported as median (IQR)

Quantitative data describing practice-setting constraints that women face when accessing cervical cancer screening are presented in Table 2. The median distance that women reported traveling to reach the closest healthcare facility was 5.5 kilometers (IQR: 3.1 - 10.1) and the median travel time to the clinic was 25.0 minutes (IQR: 15.0 - 30.0). One hundred and twenty-four women (57.7%) reported that they had to wait longer than one hour before being seen at the clinic. Most women took public transportation to get to the healthcare facility (58.1% or n = 125). Aside from public transportation, women usually took either their vehicle or a ridesharing vehicle (31.6% of n = 68). Seventy-three women (34.0%) responded that they needed permission before scheduling a doctor's appointment. Less than 40% of women knew where to obtain screening and most women (90.2% or n = 194) reported that they had never been screened for cervical cancer.

Barriers stratified by highest level of educational attainment, marital status and rural versus urban are presented in Table 3. When stratified by education level, women with no college or university were significantly more likely to have to wait >1 hour to be seen at a local clinic versus women with some college or university-level education (p = 0.001) and were significantly less likely to be aware of local clinics offering screening services (p = 0.05). When stratified by marital status, single, widowed, and divorced women were significantly more likely to walk to the clinic versus women who were married or living with their partners (p = 0.03). Married women were more likely to require permission to schedule a clinic appointment versus single women (p < 0.001). When comparing women living in an urban versus rural area, women living in an urban area spent on average, a longer duration of time traveling to their local clinic.

## Study II: Qualitative semi-structured interviews

### Materials and methods

**Study design and setting.** Key findings reported by women, related to the ecologic system and practice setting domain of the DSF, were used to guide Study II. This second phase involved conducting in-depth qualitative interviews (IDIs) with healthcare workers involved in cervical cancer screening to expand upon the survey results in Study I. For this reason, Study II was sited at two (2) urban facilities of the Ghana Health Services (GHS) CERVICARE program: Kumasi South Regional Hospital (KSRH), and Suntreso Government Hospital (SGH) and employed an explanatory descriptive qualitative design.

**Research question.** What are the potential barriers and facilitators that women encounter when accessing cervical cancer screening services at a Ghana Health Services (GHS) CERVICARE center?

**Table 2. Practice setting constraints for accessing cervical cancer screening for all women (n = 215).**

| Factor | All Women |
| --- | --- |
| How far is it, in kilometers, to the local healthcare facility/clinic from your home?* | 5.5 (3.1–10.1) |
| How much time, in minutes, does it take you to get to the clinic?* | 25.0 (15.0–30.0) |
| Wait longer than 60 minutes to be seen at the clinic, n (%) | 124 (57.7) |
| Primary mode of transportation, n (%) | |
| Car (either own or rideshare) | 68 (31.6) |
| Bicycle | 1 (.004) |
| Walking | 18 (8.4) |
| Public Transportation | 125 (58.1) |
| Needs permission to schedule a healthcare visit, n (%) | 73 (34.0) |
| Knows of clinics/hospitals that offer cervical cancer screening, n (%) | 80 (37.2) |
| Has never been screened for cervical cancer, n (%) | 194 (90.2) |
| Has access to a mobile phone, n (%) | 203 (94.4) |

*Distance to the local healthcare facility and time to clinic are reported as median (IQR)

**Table 3. Barriers stratified by education level, marital status, and rural versus urban for all women (n=215).**

| Barriers | Frequency (n) and Percentage (%) | | Prevalence Ratio | p-value |
|---|---|---|---|---|
| | No College/University (n=165) | Some College/University (n=50) | | |
| Distance (km) to local healthcare facility, mean (SD) | 8.7 (11.4) | 6.5 (6.5) | | 0.09 |
| Time (minutes) to local healthcare facility, mean (SD) | 30.8 (23.2) | 27.0 (23.3) | | 0.32 |
| >1 hour wait time at clinic %(n) | 64.8 (107) | 34.0 (17) | | <0.001 |
| Primary mode of transportation %(n) | | | | |
| Car (own or rideshare) | 24.8 (41) | 54.0 (27) | Referent | |
| Bicycle | 0.6 (1) | 0 | NA | |
| Walking | 9.7 (16) | 4.0 (2) | 0.54 | 0.41 |
| Public transportation | 63.6 (105) | 40.0 (20) | 0.78 | 0.49 |
| Needs permission to schedule visit %(n) | 32.7 (54) | 38.0 (19) | | 0.6 |
| Knows of clinics with screening %(n) | 33.3(55) | 50.0 (25) | | 0.05 |
| Never been screened for cervical cancer %(n) | 90.3 (149) | 90.0 (45) | | 1 |
| **Barriers** | **Married/Living Together (n=162), n(%)** | **Single/Divorced/Widowed (n=53), n(%)** | **Prevalence Ratio** | **p-value** |
| Distance (km) to local healthcare facility, mean (SD) | 8.4 (11.1) | 7.6 (8.2) | | 0.58 |
| Time (minutes) to local healthcare facility, mean (SD) | 30.0 (23.3) | 30.0 (23.4) | | 1 |
| >1 hour wait time at clinic %(n) | 60.5 (98) | 49.1 (26) | | 0.19 |
| Primary mode of transportation %(n) | | | | |
| Car (own or rideshare) | 32.7 (53) | 28.3 (15) | Referent | |
| Bicycle | 0 | 1.9 (1) | NA | |
| Walking | 5.6 (9) | 17.0 (9) | 2.3 (1.2–4.3) | 0.03 |
| Public transportation | 59.9 (97) | 52.8 (28) | 1.0 (0.58–1.8 | 0.96 |
| Needs permission to schedule visit %(n) | 42.6 (69) | 7.5 (4) | | <0.001 |
| Knows of clinics with screening %(n) | 40.1 (65) | 28.3 (15) | | 0.17 |
| Never been screened for cervical cancer %(n) | 88.3 (143) | 96.2 (51) | | 0.15 |
| **Facility-level barrier** | **Rural (n=70)** | **Urban (n=145)** | **Prevalence Ratio** | **p-value** |
| Distance (km) to local healthcare facility, mean (SD) | 9.2 (6.9) | 7.8 (11.8) | | 0.26 |
| Time (minutes) to local healthcare facility, mean (SD) | 24.2 (18.2) | 32.7 (24.9) | | 0.005 |
| >1 hour wait time at clinic %(n) | 61.4 (43) | 55.9 (81) | | 0.53 |
| Primary mode of transportation %(n) | | | | |
| Car (own or rideshare) | 32.9 (23) | 31.0 (45) | Referent | |
| Bicycle | 0 | 0.7 (1) | NA | |
| Walking | 5.7 (4) | 9.7 (14) | 1.18 (0.87–1.59) | 0.37 |
| Public transportation | 57.1 (40) | 58.6 (85) | 1.03 (0.83–1.27) | 0.8 |
| Needs permission to schedule visit %(n) | 31.4 (22) | 35.2 (51) | | 0.7 |
| Knows of clinics with screening %(n) | 27.1 (19) | 42.1 (61) | | 0.05 |
| Never been screened for cervical cancer %(n) | 87.1 (61) | 91.7 (133) | | 0.42 |

**Sampling procedure and sample size determination.** There was no predetermined sample size for this phase of the study. We adopted a purposive sampling technique to select eligible participants. However, by the 15th interview, there was no new analytical information, indicating that we had reached data saturation [29]. We carried out two additional interviews to confirm that we had reached data saturation resulting in a final sample size of 17.

**Recruitment and consent.** Approval for this study was provided by the Committee on Human Research, Publication and Ethics, Kwame Nkrumah University of Science and Technology, Ghana (CHRPE/AP/043/22) and the Cancer Center Protocol Review Committee, University of Minnesota, USA (CPRC# 2022LS045). For Study II, we targeted healthcare workers from each of the two urban facilities of the Ghana Health Services (GHS) CERVICARE program (KSRH and SGH) for enrollment. Healthcare workers were defined as nurses/midwives, doctors, or laboratory personnel who were involved in cervical cancer screening (nurses/midwives and doctors) or testing (laboratory personnel). Individuals were eligible for the study if they were 21 years of age or older, involved in cervical cancer screening, could comprehend the questionnaire verbally and could provide informed consent. Possible participants were identified by administrators who volunteered to serve as "recruitment links" at each selected facility. A trained research assistant (RA) met each of the possible participants to assess eligibility and invite them to participate in the study. Specifically, the RA used a study information sheet to provide details, including the goals of the study, what is required of each participant, and a reminder that participation was voluntary. Potential participants were then given 1 week to consider participation. The interviews took place in a private office at each facility. Oral and written consent were obtained from each participant before the interview. The interviews were carried out during November 2022.

**Data collection tool and procedure.** A semi-structured interview guide was used. This guide was composed of two sections, with the first section asking for demographic information such as age, gender, education level, and length of employment at the current location. Key findings from the quantitative data (such as needing permission to schedule a clinic visit, knowledge about where to obtain screening, long wait time at the clinic and the distance to health facilities that were aligned with the ecological system and practice setting domains of the DSF shaped the interview questions that were asked of healthcare workers in the second section. Two research assistants conducted all 17 interviews. Each interview was conducted face-to-face and was audio recorded. The interviews were conducted in English (n = 4) and local language (Twi, n = 13) and lasted 30 minutes on average.

**Data analysis.** A deductive thematic analysis framework was used [30]. The process of analysis began with verbatim transcription of both the English and Twi audio-recorded data. For the interviews conducted in Twi, two expert translators were used in a back-back translation process. A random selection of audio data was evaluated by a bilingual research assistant to ensure accuracy in interviewing, transcribing, and translation. Transcripts were imported into QSR NVivo-12 Plus for data management and coding. The 'nodes' function in QSR NVivo-12 was used for preliminary inductive coding [31]. We had a list of codes ("start list") based on the practice setting [20], but we also created other codes that emerged from the data and were not part of the start list. Two analysts (A.B.B.M and J.O) independently coded the transcripts in three stages. For the first stage, they independently assigned codes to text sections based on the ecological system and practice setting domains of the DSF. For the second stage, they independently re-reviewed the data to identify emerging codes that were not part of the domains of the DSF. They then discussed the codes, clarified discrepancies, revised definitions, and created new codes. For the third stage, the analysts jointly organized the codes into themes and sub-themes guided by the domains of the DSF [20]. The deductive coding results were categorized according to emerging patterns and major themes around barriers and facilitators to cervical cancer screening and testing. Key quotes were extracted to reflect the various themes related to barriers and facilitators. The inter-coder agreement was 95%. To enhance the trustworthiness of the data, strategies such as prolonged engagement, peer debriefing, member checking (n = 2), and detailed descriptions of the methodological processes and context were employed [32]. The in-depth analysis process further ensured the reliability of the results [33].

## Results

### Demographic characteristics

The demographic characteristics for the healthcare workers' sample are presented in Table 4. The median age of healthcare workers was 38.0 years (IQR: 36.5 - 43.5). Most healthcare workers identified as female (78.9%, n = 15) and reported a university degree (63.2%, n = 12). The median length of time that healthcare workers had worked within their profession was 12.0 years (IQR: 10.0 - 15.5).

### Themes

Two main themes related to the practice setting emerged from the data through thematic analysis: client-level barriers and health system challenges. Each theme had sub-themes (Table 5).

Healthcare workers identified several client-level barriers, reflecting characteristics of the patient population served by the CERVICARE program. One key barrier was that women perceived financial constraints to cervical cancer screening either through the direct cost of screening services or the indirect cost associated with transportation to the clinic. Healthcare workers also noted that women often do not think the financial burden of screening services is worth the benefit it provides in detecting cervical cancer:

> "Most of them complain about finances. They think that why should I use my money to go and do screening while I can use it for other pressing matters. So, it is not seen as a priority" (P00 2, laboratory personnel)

**Table 4. Demographics of healthcare workers.**

| Participant Characteristic | All healthcare workers (n = 19) |
|---|---|
| Age in years[*] | 38.0 (36.5–43.5) |
| Gender, n (%) | |
| Female | 15 (78.9) |
| Male | 4 (21.1) |
| Education, n (%) | |
| Some college | 7 (36.8) |
| A college/university degree | 12 (63.2) |
| Employment | |
| Length of employment in years[*] | 12.0 (10.0–15.5) |

[*]Age and length of employment are presented in median (IQR)

**Table 5. Themes and subthemes from healthcare worker interviews, aligned with DSF domains.**

| DSF Domain | Themes | Sub-themes |
|---|---|---|
| Ecological system – population characteristics | Client level barriers | Financial constraints and cost of treatment |
| | | Clients' non-adherence due to shyness and feelings of discomfort |
| | | Fatalistic views about the outcome of cervical cancer screening |
| Practice Setting | Health system challenges | Infrastructure inadequacies |
| | | Logistical constraints |
| | | Inadequate staffing of clinics |

Healthcare workers also discussed non-adherence with screening due to feelings of discomfort around the VIA procedure. Women experienced shyness for several reasons, including fear of judgment on personal hygiene or discomfort from healthcare workers seeing them naked:

"Some women are shy of themselves for their fellow women to see their nakedness…. this makes them non-compliant and difficult to handle. At the end of the day, you will not be able to do the screening because they will be reluctant to follow the screening procedures." (P006, midwife)

Fears about the results from the screening process was another theme that emerged from the healthcare workers' interviews. According to the healthcare workers, their clients feared that the screening would reveal that they had the disease (cervical cancer). This fear about a positive screening result was attributed to fatalistic views held by the clients. That is, healthcare workers thought that clients feared that once they had been diagnosed as having cervical cancer, it would mean that they were going to die. Consequently, this fear made the clients non-adherent and unwilling to get screened.

"Fear of the unknown. You know, after educating the women and giving them counseling, they will still not feel comfortable for you to screen them because of fear of the unknown. She doesn't know what the outcome will be. The fear of being diagnosed with cervical cancer and dying from the disease." (P00 14, midwife)

The overarching theme of practice setting constraints, "health system challenges," also had three sub-themes: infrastructure inadequacies, logical constraints, and inadequate staffing of clinics. Healthcare workers noted that the infrastructure capacity of healthcare facilities was not adequate for providing women's health services. For example, clinics lacked the beds necessary for cervical cancer screening, or the clinic setup did not allow for patient privacy. In clinics with a proper examination room, patients often experienced long wait times due to the availability of only one or a few rooms:

"We don't have a specific room with a special bed. The room that we use is not ideal; it does not support the privacy of the client in any way. Because of that, the clients don't feel comfortable getting screened." (P001, midwife)

Lastly, healthcare workers noted that inadequate staffing of clinics resulted in longer wait times. This limits patients from returning for future visits, and patients may also advise others against wanting to use the clinic's services:

"Inadequate staffing is the big issue. We are few, hence, the women wait for a long time to be attended to. They go and do not come back. Sharing such experience with other women also discourages screening uptake." (P0016, midwife)

## Discussion

In this study, we adopted domains of the DSF, namely the practice setting and ecological system, to characterize cervical cancer screening in Ghana and identify constraints that need to be addressed to increase screening uptake and improve outcomes. In terms of factors that affect screening at the ecological system level, the quantitative survey indicated that only 37.2% of women were aware of facilities offering screening services. This finding is consistent with studies conducted in Uganda [34] and Tanzania [35]. An earlier (2016) study in Uganda found that less than half of surveyed participants were aware of cervical cancer screening services [34], and a 2012 study in Tanzania found that women with higher levels of cervical cancer prevention knowledge were significantly more likely to partake in screening services [35]. This lack of awareness about cervical cancer prevention may be attributable to the lack of a national cervical cancer screening program, which would provide women with an organized, guideline-based screening that's advertised and widely adopted, as well as the lack of coordinated outreach for the few clinics that do offer screening. Efforts to increase awareness about cervical cancer including screening through educational campaigns, could address this.

In terms of population characteristics that can impact the uptake of screening, we found that women with lower educational attainment were less likely to be aware of facilities that provide cervical cancer screening services. Despite Ghana's progress in education, illiteracy remains a widespread issue, particularly among women, the poor, and rural populations [21]. Approximately 42% of adults are illiterate, with women (50%) being more affected than men (33%) [21] This challenge is magnified by the reliance on print media for information, limiting women's ability to make informed health decisions [36,37]. This finding highlights the need to consider a wider array of outreach options, beyond printed literature to inform women about cervical cancer screening that accounts for educational levels, such as radio, television, and community center announcements.

Another population characteristic that we noted in our study that can impact screening and is consistent with other studies conducted in LMICs such as Pakistan (36), Malaysia [38], and India [39,40] is that some women reported needing permission to schedule appointments. The study revealed that married women were more likely to need permission before they could schedule a visit. This finding may be explained by existing patriarchal and gender norms that place the male as the decision-maker in the household [41,42]. Consequently, the autonomy of women to make decisions regarding their healthcare is significantly reduced, especially among married women. Another consideration is Ghanaian traditional culture, which encourages women to ensure that their nakedness is only seen by their husbands or partners. Given that the cervix is examined during cervical cancer screening, married women may feel uncomfortable undergoing this process [12]. Those willing to undergo screening may require their partners to grant permission before they can allow another person to see their 'nakedness.' This assertion was corroborated by the interview accounts of the healthcare workers regarding the shyness and discomfort women feel concerning cervical cancer screening. These results suggest that to increase screening uptake, other options, such as HPV test-based screening through self-collected samples, should be considered since self-collection would not require the assistance of healthcare providers and could ensure privacy [43]. Previous studies in Ghana and other sub-Saharan African countries such as Tanzania have shown that women feel comfortable and confident with this type of screening [44,45].

In terms of the practice setting in which screening is offered, contrary to previous studies that have found long travel times for cervical cancer screening among women in rural areas [39,40], we found that longer travel times were more likely to be experienced by those living in urban areas. This is possibly due to higher population density, traffic congestion, or longer distances between residences and healthcare facilities in urban areas compared to rural areas. Further research is required to fully comprehend why urban-dwelling women in Ghana spend long travel time in accessing cervical cancer screening services.

Another key consideration that reflects the context in which screening is offered is the long wait time, which is a major barrier to accessing cervical cancer screening, with more than half of participants reporting waiting over 1 hour to be seen at the clinic. This result is consistent with a study in which 86% of a sample of 200 Kenyan women reported long wait times as a barrier to cervical cancer screening participation [46]. Similar findings were also reported in a qualitative study of 48 women conducted in Accra, Ghana [47]. Of note, our study also found that women with lower educational attainment were more likely to experience longer wait times at local clinics. A plausible explanation for this finding could be that women with higher education are more likely to access cervical cancer screening services in private healthcare facilities with less congestion and faster service delivery. This study did find that women with higher educational attainment reported on average, higher monthly incomes, which may also allow them to access more expensive, private clinics. Women with lower educational attainment may be more likely to access cervical cancer screening services at public healthcare facilities where there is a high volume of patients, long queues, and fewer staff and/or rooms available for screening [48]. This finding was confirmed by interviews with healthcare workers who noted that low staffing in their facilities resulted in longer wait times, which, in turn, discouraged women from accessing cervical cancer screening services. Another perspective is that lower educational attainment may be a proxy for socioeconomic status and lower information attainment. Women with lower levels of education often face limited access to economic opportunities, resulting in lower income levels and resources. This economic disadvantage can directly impact their ability to access healthcare services,

including cervical cancer screening, due to financial constraints such as transportation costs or the inability to afford screening fees. While the long wait times and staffing could be addressed with a switch to screening using self-collected HPV tests, the lack of awareness and costs still need to be addressed to increase uptake and improve the effectiveness of cervical cancer screening as currently offered.

## Strengths and limitations

The use of a mixed methods approach is one of the strengths of our study as the qualitative component provides more explanation to the nuances observed in the survey. Another strength of this study is application of the DSF to examine why established cervical cancer screening interventions are not reaching women and pinpoint critical aspects from both the practice setting and ecological system domains that need to be addressed to increase participation. The stratification of barriers by various demographic characteristics helps provide insight as to which sub-populations of women are likely to experience specific barriers, which can inform tailored interventions. Nonetheless, there are some noteworthy limitations to our study. The use of convenience sampling may have resulted in a selected sample that limits the generalizability of the findings from this study to the larger population of women in the Ashanti region. Given that we relied on administrators at the healthcare facilities to assist with the identification of potential healthcare worker participants, there is the possibility of selection bias. However, we tried to minimize this risk of bias by conducting a detailed eligibility screening. Also, limiting the study to women who only visited the three facilities precludes our obtaining information on the barriers faced by women who do not visit these facilities. We also acknowledge that, while the quantitative survey was adapted from a pre-tested survey, the quantitative survey was not assessed for face and content validity. This study did not account for the time component of the DSF, limiting our ability to assess how changes over time may have influenced the identified barriers or informed future intervention planning. Finally, limiting the in-depth interviews to only healthcare workers does not comprehensively explain the health system challenges experienced by the women themselves or allow for other perspectives that can influence the success of screening, such as those of national and local policymakers.

## Conclusion

In conclusion, this study finds that a large proportion of Ghanaian women report never having been screened for cervical cancer. Both the quantitative and qualitative studies highlight barriers that women face accessing cervical cancer screening. The women's survey found that long clinic wait times may impact screening uptake. This was especially true for women with lower educational attainment (i.e., those with no college or university education). Additionally, there was an overall lack of knowledge regarding where to obtain screening services. Of note, one-third of surveyed women (and over two-thirds of surveyed married women) needed permission before scheduling a doctor appointment. In-depth interviews with healthcare workers confirmed that healthcare facilities were not adequately staffed, resulting in longer patient wait times. Healthcare workers also noted culturally sensitive issues as a barrier to screening uptake including patient shyness. The use of self-collected samples for HPV testing could mitigate barriers such as time spent traveling to and waiting at a healthcare clinic. It might also provide a more culturally sensitive option for screening to address cultural norms since it does not require the assistance of healthcare providers. Future studies should assess the feasibility of implementing self-collected HPV samples as a possible method for cervical cancer screening.

## Acknowledgments

We are grateful to all participants who shared their experiences in this study.

## Author contributions

**Conceptualization:** Adwoa Bemah Boamah Mensah, Thomas Okpoti Konney, Beth Virnig, Shalini Kulasingam.

**Data curation:** Adwoa Bemah Boamah Mensah, Thomas Okpoti Konney, Madalyn Nones, Joshua Okyere, Beth Virnig, Shalini Kulasingam.

**Formal analysis:** Adwoa Bemah Boamah Mensah, Madalyn Nones, Joshua Okyere, Shalini Kulasingam.

**Funding acquisition:** Adwoa Bemah Boamah Mensah, Thomas Okpoti Konney, Beth Virnig, Shalini Kulasingam.

**Investigation:** Adwoa Bemah Boamah Mensah, Thomas Okpoti Konney, Ernest Adankwah, John Amuasi, Madalyn Nones, Kwame Ofori Boadu, Felicia Maame Efua Eduah, Beth Virnig, Shalini Kulasingam.

**Methodology:** Adwoa Bemah Boamah Mensah, Thomas Okpoti Konney, Ernest Adankwah, John Amuasi, Madalyn Nones, Kwame Ofori Boadu, Felicia Maame Efua Eduah, Serena Xiong, Jeong Robin Moon, Beth Virnig, Shalini Kulasingam.

**Project administration:** Adwoa Bemah Boamah Mensah, Shalini Kulasingam.

**Validation:** Adwoa Bemah Boamah Mensah, Shalini Kulasingam.

**Writing – original draft:** Adwoa Bemah Boamah Mensah, Madalyn Nones, Joshua Okyere, Shalini Kulasingam.

**Writing – review & editing:** Adwoa Bemah Boamah Mensah, Thomas Okpoti Konney, Ernest Adankwah, John Amuasi, Madalyn Nones, Joshua Okyere, Kwame Ofori Boadu, Felicia Maame Efua Eduah, Serena Xiong, Jeong Robin Moon, Beth Virnig, Shalini Kulasingam.

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
