## [Decision Letter · Decision Letter 0]

29 May 2024

PGPH-D-24-00360

Perceptions of barriers and facilitators for cervical cancer screening from women and healthcare workers in Ghana: Applying the Dynamic Sustainability Framework.

Dear Dr. BOAMAH MENSAH,

Thank you for submitting your manuscript to PLOS Global Public Health. After careful consideration, we feel that it has merit but does not fully meet PLOS Global Public Health’s publication criteria as it currently stands. Therefore, we invite you to submit a revised version of the manuscript that addresses the points raised during the review process.

 Reviewer 2 has done an extensive review and the comments may kindly be addressed in detail.

There are certain methodological issues that need to be addressed particularly with respect to the qualitative research design. This requires a major revision.

We look forward to receiving your revised manuscript.

Kind regards,

Suma Nair

Academic Editor

Journal Requirements:

Additional Editor Comments (if provided):

Reviewer reports are attached for your refernce. There are certain methodological issues that need to be addressed particularly with respect to the qualitative research design. This requires a major revision.

Reviewers' comments:

Reviewer's Responses to Questions

**Comments to the Author**

1. Does this manuscript meet PLOS Global Public Health’s publication criteria?

Reviewer #1: Yes

Reviewer #2: No

2. Has the statistical analysis been performed appropriately and rigorously?

Reviewer #1: Yes

Reviewer #2: Yes

3. Have the authors made all data underlying the findings in their manuscript fully available (please refer to the Data Availability Statement at the start of the manuscript PDF file)?

Reviewer #1: No

Reviewer #2: No

4. Is the manuscript presented in an intelligible fashion and written in standard English?

Reviewer #1: Yes

Reviewer #2: No

Reviewer #1: The manuscript would benefit from expanded recommendations for policy and practice, particularly detailing the operationalization of suggested interventions like self-collected HPV tests within the Ghanaian healthcare system. Addressing the limitations more comprehensively, particularly regarding sampling strategy and potential biases, along with a clearer outline of future research directions, would strengthen the manuscript. Minor editorial revisions for clarity and coherence are recommended to polish the presentation, please refer to the comments in the PDF. Addressing these areas can amplify the study's contribution to public health efforts in combating cervical cancer in low-resource settings.

Reviewer #2: Perceptions of barriers and facilitators for cervical cancer screening from women and healthcare workers in Ghana: Applying the Dynamic Sustainability Framework.

Manuscript reference number: PGPH-D-24-00360

1. Original submission – research paper

1.1 Recommendation

Major revision

2. Comments to the Authors

2.1 Overview

Congratulations to the authors for conducting this study on understanding barriers and facilitators to cervical cancer screening for women in Ghana. Certainly, in low- and middle-income countries and, especially those of sub-Sahara Africa, notwithstanding issues of patient and social factors such as education and health literacy, gender and culture, limited availability of screening technologies and facilities serves to compound rates of uptake. This manuscript reflects an important and prevalent situation in many low- and middle-income countries and will be of interest to the readership of PLOS Global Public Health.

2.2 Major comments

The authors state that the study is designed in line with an exploratory, mixed-methods sequential approach. While I understand the author’s intentions to conduct a study of this nature, there are two overarching comments I would like to make and these centre on the author’s use of the Dynamic Sustainability Framework (DSF) in this study and the conceptualisation of the way in which this study is presented in this manuscript.

The explication by the authors of the application of the DSF to this study is very lacking, indeed, I would argue, almost absent in this manuscript. At a minimum, the reporting of the study is confusing and unclear. As such, there are areas in the paper that warrant additional explanation, particularly parts of the research process and, of course, the conceptual framework of the DSF. I explain this in my following comments.

2.2.1 Dynamic Sustainability Framework

The authors have used the DSF in this study as a conceptual framework. They have taken components of the model to inform the study design and drive the data collection process. Whilst this is not incorrect, it is completely unclear to the reader how the authors have arrived at any decision making on this use of the DSF in this study. I suspect this is why the authors state that they “…used the general DSF in combination with a mixed methods approach…”. I argue this is fundamentally incorrect. The DSF is the conceptual framework of this study and should be applied and explained in this manner. To this end, what do the authors mean by “the general DSF”? Is this different from the DSF reported in various literature?

Moreover, in addition to unclarity as to what is the “general DSF”, the authors use “ecological system” to guide the survey work and “practice setting” to guide the qualitative work. How or why was it decided to “break up” up the DSF model in this way? What was the decision making for seemingly “plucking out” these domains of the model? Is it because existing screening technologies and facilities in Ghana are considered to already fit within the “Intervention” domain and therefore the focus of this study is to unpack issues in the “practice setting/context” and “ecological system”? I think a large part of my confusion around the authors use of the DSF in this study is that Figure 1 is missing the critical detail of “practice setting”. Ensuring Figure 1 is accurate will go a long way to helping readers understand this better.

2.2.2. Reporting

In view of the confusion that this paper generates for the reader, particularly in regard to the use of the DSF, it is my suggestion that the authors consider adding in a new section named “Conceptual Framework” and spending time to more deeply conceptualise the use of the DSF to this study. Further, the DSF in a new section can be better explained, including what the domains are all about and how they apply to this type and focus of study, for example, to the uninitiated reader, there is no explanation of what “ecological system” theoretically means. In so doing, the authors can then give time to explaining how such constructs informed the approach to data collection. Currently, there is no reference to the application of the constructs to the data collection in either the quant or qual study.

On the point of reporting, this is a challenging paper to read. There are sections of the research process, for example, eligibility (line 186), recruitment (sampling approach) (line189) and consent (line 193) are reported in the data collection section. Data collection should only focus on the instrument used. Another example of the challenge that this paper presents to the reader is the reporting of results of the quant survey after the qualitative data collection section – this makes it difficult to decipher how the findings of the quant survey were developed to inform the qual data collection/interview guide.

It is my suggestion that the authors might like to reconsider restructuring the reporting of the study as two distinct studies, to enhance clarity throughout. For instance, study 1 is the quant study and would align with standard reporting criteria – design and setting, sample and sampling, recruitment and consent, data collection, data analysis, results. The introduction of study 2 will therefore be better positioned, clearer and logically placed in the paper and can be reported similarly – design and setting, sample and sampling, recruitment and consent, data collection, data analysis, results. A discussion section can then follow that discusses the overall findings and, important, is discussed in view of the principles of the DSF – which, currently, is lacking in the discussion.

Such an approach to restructuring the paper would also help prevent what appears to be a randomly placed section in the paper on “Ethics approval and consent process”, when the consent processes have already been previously discussed (albeit incorrectly) in the quant and qual data collection sections.

2.3 General comments

2.3.1 Quant survey study

Line 148 states that “individuals were selected” to participate. Is this correct? How exactly were they recruited? Was there a decision making process on which women were approached? Is it possible that the ‘participant information and consent sheet’ used in this study could be included as supplementary material?

Was a sample size calculated for this first phase of the study? There is no mention of this in the paper.

The issue of bias is not addressed – and, on this note, a strengths and limitations section is not included in the paper. Bias might have been an issue as women were recruited from centres. This assumes only those women able to attend centres were included in the study. Moreover, the issue of payment made to participants is not addressed as a potentially biasing factor.

More information is needed on the tool used, for example, how many factors and items did it originally start with? Was it a validated tool in the Ugandan study? What part of the original tool was adapted? After adaption, was face and content validity tested before dissemination to the participants?

2.3.2 From quantitative to quantitative methods

How was the importance of particular quantitative findings decided over others in terms of informing what would be asked in the interviews?

Why was it decided to only interview health workers perceptions on health system challenges experienced by women? Having not interviewed women themselves seems to be a loss in this study.

Quantitative findings are reported in this short section before the formal Results section.

2.3.3 Qualitative study

The authors state that the administrators identified potential participants – based on what criteria? Bias in this process will need to be addressed.

Results of the qual study (e.g., sample size of 17) is reported in amongst the eligibility, recruitment and consent information (which supports my earlier point about restructuring this paper to report it in line with standard reporting criteria). Data analysis is also reported in this section, for example, information about how theoretical saturation was arrived at should be in a dedicated data analysis section.

2.3.4 Results

Line 329 – “based on the results of the quantitative analysis, two main themes related to the practice setting were further explored….” – how were these themes arrived at from the quant data to know that they should be explored in the interviews?? Was there additional theoretically driven analysis that occurred with the quant data, other than the descriptive and analytical statistical approaches stated?

Did the authors also prepare an EQUATOR study report document with their submission to PGPH?

**Do you want your identity to be public for this peer review?** For information about this choice, including consent withdrawal, please see our Privacy Policy

Reviewer #1: **Yes: ** Khushbu Balsara

Reviewer #2: No

---

## [Decision Letter · Decision Letter 1]

27 Dec 2024

PGPH-D-24-00360R1

Perceptions of barriers and facilitators for cervical cancer screening from women and healthcare workers in Ghana: Applying the Dynamic Sustainability Framework

Dear Dr. BOAMAH MENSAH,

Thank you for submitting your manuscript to PLOS Global Public Health. After careful consideration, we feel that it has merit but does not fully meet PLOS Global Public Health’s publication criteria as it currently stands. Therefore, we invite you to submit a revised version of the manuscript that addresses the points raised during the review process.

We look forward to receiving your revised manuscript.

Kind regards,

Edina Amponsah-Dacosta, Ph.D., MPH

Academic Editor

Additional Editor Comments (if provided):

Reviewers' comments:

Reviewer's Responses to Questions

**Comments to the Author**

Reviewer #1: All comments have been addressed

Reviewer #2: (No Response)

publication criteria?

Reviewer #1: Yes

Reviewer #2: Partly

3. Has the statistical analysis been performed appropriately and rigorously?

Reviewer #1: (No Response)

Reviewer #2: No

4. Have the authors made all data underlying the findings in their manuscript fully available (please refer to the Data Availability Statement at the start of the manuscript PDF file)?

Reviewer #1: Yes

Reviewer #2: (No Response)

5. Is the manuscript presented in an intelligible fashion and written in standard English?

Reviewer #1: Yes

Reviewer #2: Yes

Reviewer #1: My previous comments have been addressed

Reviewer #2: Please see my document attached with full comments.

**Do you want your identity to be public for this peer review?** For information about this choice, including consent withdrawal, please see our Privacy Policy

Reviewer #1: No

Reviewer #2: No

---

## [Editor Report · Decision Letter 2]

9 Apr 2025

Perceptions of barriers and facilitators for cervical cancer screening from women and healthcare workers in Ghana: Applying the Dynamic Sustainability Framework

PGPH-D-24-00360R2

Dear Dr BOAMAH MENSAH,

We are pleased to inform you that your manuscript 'Perceptions of barriers and facilitators for cervical cancer screening from women and healthcare workers in Ghana: Applying the Dynamic Sustainability Framework' has been provisionally accepted for publication in PLOS Global Public Health.

Best regards,

Edina Amponsah-Dacosta, Ph.D., MPH

Academic Editor